# A Circularly Polarized mmWave Dielectric-Resonator-Antenna Array for Off-Body Communications

Tarek S. Abdou *, Rola Saad * and Salam K. Khamas *

Communications Research Group, Department of Electronic and Electrical Engineering,
The University of Sheffield, Mapping Street, Sheffield S1 3JD, UK
* Correspondence: tsabdou1@sheffield.ac.uk (T.S.A.); r.saad@sheffield.ac.uk (R.S.);
  s.khamas@sheffield.ac.uk (S.K.K.)

**Abstract:** This paper presents a novel 28 GHz circularly polarized rectangular dielectric-resonator antenna (DRA) array for millimeter-wave (mmWave) off-body applications. A feed network incorporating cross slots was utilized in the rectangular DRA design to realize circular polarization (CP). In terms of the free-space wavelength, $\lambda_o$, the DRA dimensions were ($0.48\lambda_o \times 0.48\lambda_o \times 0.27\lambda_o$) at 28 GHz. The antenna array was simulated by incorporating dielectric layers with parameters that are equivalent to those of the human body at the desired frequency of 28 GHz for off-body communications. Moreover, the precise alignment and assembly of the DRA, which pose major difficulties at mmWave frequencies, were achieved by outlining the DRA positions on the ground plane using a three-dimensional (3D) printer. The array configuration was fabricated and measured with excellent performance, realizing a measured impedance bandwidth of 29% in conjunction with an axial-ration (AR) bandwidth of 13% and a broadside gain of 13.7 dBic at 28 GHz.

**Keywords:** millimeter waves; antenna array; dielectric-resonator antenna; off-body communications; circular polarization

## 1. Introduction

Dielectric-resonator antennas have undergone extensive research over recent decades [1,2]. Dielectric resonators were first delivered in microwave circuits as oscillators or filters, replacing steel waveguide cavities owing to their sizable advantages. When mounted on a metal ground plane and excited properly, they radiate energy effectively and function as antennas [3]. In addition to their low profile and light weight, DRAs also have a number of other appealing qualities, such as the absence of metal and surface-wave losses [4]. Therefore, DRAs are ideal candidates for wireless communication systems, especially those that operate in the mmWave band. Traditional microstrip antennas, on the other hand, exhibit substantial surface-wave and ohmic losses at high frequencies, which drastically deteriorate the radiation efficiency [5,6].

Moreover, there are many studies on microstrip antennas that are focused on body-centric networks. However, in general, microstrip antennas have narrow bandwidths and need to be mounted on a part of the body that is less vulnerable to bending and wrinkling [7]. Hence, DRA has been proposed as an alternative option [8]. Furthermore, it is well known that CP offers a flexible orientation between the transmitting and receiving antennas compared to linear polarization (LP). Several studies have reported on CP radiating elements, such as printed antennas [9] as well as sequentially rotated CP DRA arrays [10]. Furthermore, G-shaped monopoles were utilized in a multiple-input-multiple-output (MIMO) antenna with wideband impedance and CP bandwidths [11]. In [12] a dual polarized, multiband four-port decagon-shaped flexible MIMO antenna was proposed. Another MIMO antenna with wideband impedance and circular-polarization bandwidths was proposed in [13]. Moreover, various publications have examined the effectiveness of cross-slot feeding to achieve CP radiation by exciting two degenerate DRA resonance modes with equal amplitudes and 90° phase difference with a wide CP bandwidth [14–16].

A number of research studies have utilized mmWave DRA arrays. For example, a 2 × 2 DRA array was fabricated using substrate-integrated technology with a gain of 12.7 dBic, together with impedance and CP bandwidths of 16.4% and 1.1%, respectively [5]. An aperture-coupled 2 × 2 DRA array was reported in which each cylindrical DRA element operated in the $HEM_{11\delta}$ resonance mode, which led to a gain of 11.43 dBi, as well as an impedance bandwidth of 26% [17]. In addition, a monolithic polymer-based DRA array was proposed that used four elements with a realized broadside gain of 10.5 dBi and stable radiation patterns at 60 GHz, along with an impedance bandwidth of 12% [18]. Another study proposed a 1 × 8-element-grid dielectric resonator antenna array with a measured impedance bandwidth of 18.3% and a broadside gain of 12 dBi [19]. Moreover, a 2 × 2 circularly polarized DRA array with an operating frequency of 30 GHz and a maximum gain of 9.5 dBic with impedance and axial-ratio bandwidths of 33% and 5%, respectively, was reported [20]. In order to easily align the individual array elements, a grooved and grounded superstrate was utilized for a 16-cylinder DRA array with a measured impedance bandwidth of 9.8% and a maximum gain of 15.68 dBi at 28.72 GHz [21]. Furthermore, it was demonstrated that when the DRA elements are joined using dielectric arms, for alignment purposes, the achieved impedance bandwidth and gain are 31% and 9.8 dBi, respectively [22]. From the literature, it can be noted that the highest reported gain of a mmWave DRA array is 15.68 dBi; this was achieved by using 16 linearly polarized DRA elements [21]. However, the increased number of elements is associated with a larger antenna size, as well as a complex feed network. Therefore, arrays of four circularly polarized DRA elements address these limitations, albeit with a lower gain [5,17,20]. As a result, a compact array with the advantages of higher gain and wider CP bandwidth would be beneficial and was not reported earlier at the considered frequency range.

In this study, a compact rectangular DRA element that operates at higher-order resonance modes was utilized to design a 2 × 2-element array. Compared to the research described above, the proposed DRA array offers the advantages of compact size, a wide impedance bandwidth of 29% combined with a wide AR bandwidth of 13% and an enhanced gain of 13.7 dBic at 28 GHz. Moreover, in order to overcome the misalignment issues, an automated process was followed to precisely outline the DRA positions on the utilized ground plane. The simulations were conducted using CST Microwave Studio, with close agreement between the measured and simulated results.

## 2. Design Methodology

A single DRA element was developed first to operate at 28 GHz. The parameters of the proposed DRA array were set based on mmWave off-body-communication requirements, such as low profile, wideband and high gain. The proposed antenna was simulated in free space as well as on top of a three-layer model of human tissue to investigate the effects of the human body on performance.

### 2.1. A Single DRA Element

Figure 1 illustrates the antenna's configuration, with dielectric constants and loss tangents of $\varepsilon_r$ = 9.9, tanδ < 0.0001 and $\varepsilon_s$ = 3.66, tanδ < 0.0027 for the DRA and substrate, respectively. The DRA dimensions were determined using the dielectric waveguide model (DWM), as well as a CST eigenmode solver for calculating the resonance frequencies of the transverse electric, $TE_{mnp}$, resonance modes [23].

In order to achieve the CP radiation, the single DRA element was excited by a cross slot while the energy coupling was controlled by a stub-length extension ($l_{stub}$) with a recommended initial value of $0.25\lambda_g$, where $\lambda_g$ is the guided wavelength. The cross-slot was fed using a 50-ohm microstrip line. The choice of cross-slot arm lengths and widths is crucial for maximizing the impedance and AR bandwidths [24]. The lengths of the 1st and 2nd cross-slot arms were optimized as 1.7 mm and 3.2 mm, respectively, with an identical width of $w_s$ = 0.35 mm for each arm. The antenna parameters are listed in Table 1. In addition, the higher-order modes of $TE_{113}$ and $TE_{133}$ were simultaneously excited at

27 GHz and 30 GHz, respectively, together with the fundamental resonance mode of $TE_{111}$ at 25 GHz. Table 2 presents the resonance frequencies of the transverse electric modes, which are supported by the designed DRA within the considered frequency range.

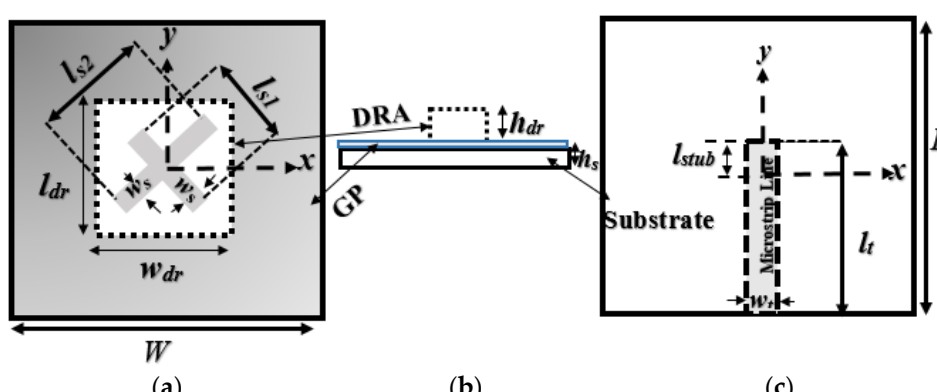

**Figure 1.** Antenna configuration: (**a**) top view, (**b**) front view, (**c**) back view.

**Table 1.** Parameters of the proposed single DRA element.

| Parameters | Value (mm) | Description |
|---|---|---|
| $w_{dr}$ | 5.2 | DRA width |
| $l_{dr}$ | 5.2 | DRA length |
| $h_{dr}$ | 2.9 | DRA height |
| $L$ | 19 | Length of ground plane |
| $W$ | 21 | Width of ground plane |
| $h_s$ | 0.254 | Substrate height |
| $l_t$ | 10.5 | Length of microstrip line |
| $w_t$ | 0.45 | Width of microstrip line |
| $l_{stub}$ | 1 | Stub length |
| $l_{s1}$ | 1.7 | Length of the first slot |
| $l_{s2}$ | 3.2 | Length of the second slot |
| $w_s$ | 0.35 | Width of the cross slot |

**Table 2.** Resonance frequencies of a number of TE modes supported by the DRA.

| Frequency (GHz) | Resonance Mode |
|---|---|
| 25 | $TE_{111}$ |
| 27 | $TE_{113}$ |
| 30 | $TE_{133}$ |

### 2.2. Single DRA Next to a Human Body

To investigate the impact of human body on the antenna performance, the DRA was mounted on a three-layer tissue model. The modeled layers of the human-body tissues, measuring $100 \times 45 \times 13$ mm$^3$, are illustrated in Figure 2. Separation distances of 1 mm and 5 mm between the DRA and the body phantom were considered [25]. The characterizations of the body parameters were used according to those given in [26], as summarized in Table 3.

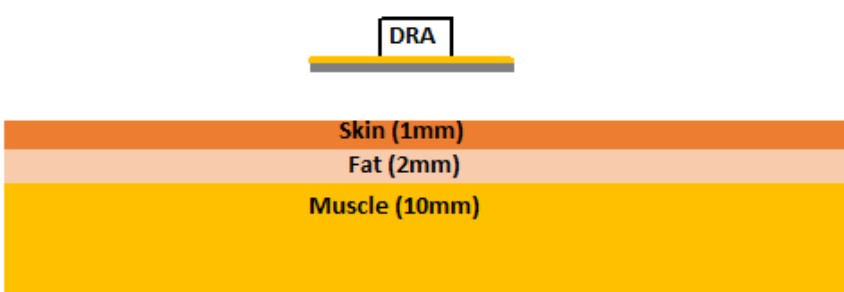

**Figure 2.** A single DRA element above the body phantom.

**Table 3.** Human-body-tissue parameters at 28 GHz.

| Tissue | Skin | Fat | Muscle |
|---|---|---|---|
| Relative Permittivity | 16.55 | 6.09 | 25.43 |
| Density (kg/m$^3$) | 1109 | 911 | 1090 |
| Conductivity (S/m) | 25.82 | 5.04 | 33.6 |
| Thickness (mm) | 1 | 2 | 10 |

Figure 3 illustrates the reflection coefficient when the DRA is positioned in free space, as well as at distances of 1 mm and 5 mm above the phantom. For both distances, the achieved impedance bandwidth is ~31%, which is close to that of ~31.5% for a DRA in free space. This demonstrates the marginal impact of the human tissues on the impedance bandwidth, which can be explained by the presence of a ground plane that minimizes the interaction between the DRA and tissues [14,27].

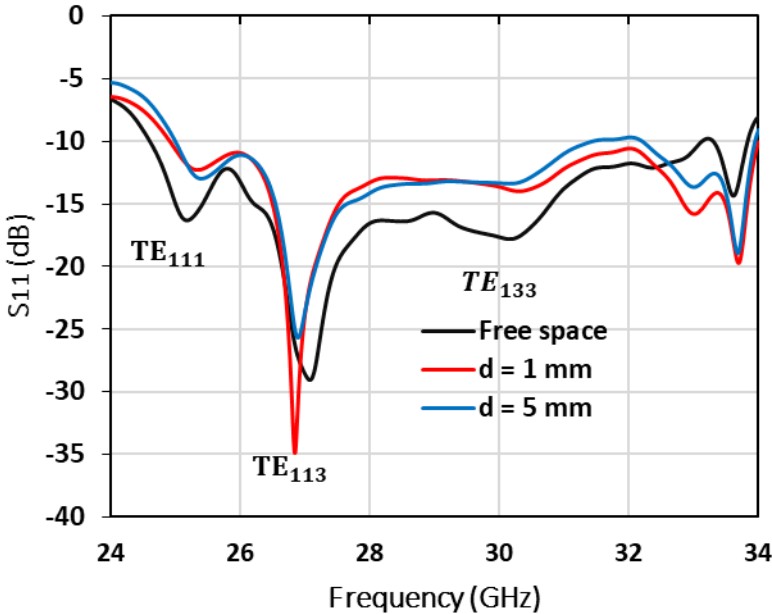

**Figure 3.** |S$_{11}$| of a DRA in free space and placed next to a phantom.

The same was also confirmed for the circularly polarized radiation. As shown in Figure 4, almost the same axial-ratio bandwidth, of 14%, was achieved in free space, as well at various separation distances between the antenna and the phantom. This was also the case for the achieved gain of ~8.7 dBic at 28 GHz. The gain is illustrated in Figure 5.

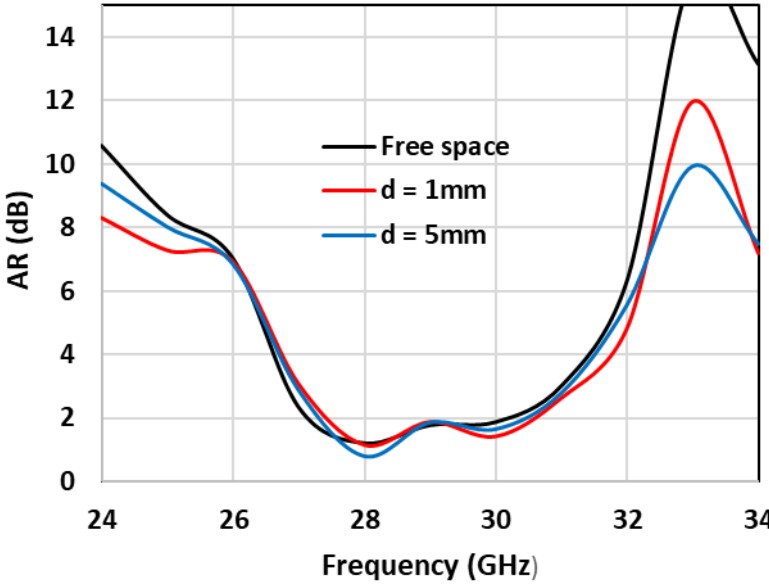

**Figure 4.** Axial ratio of a DRA placed in free space and next to a phantom.

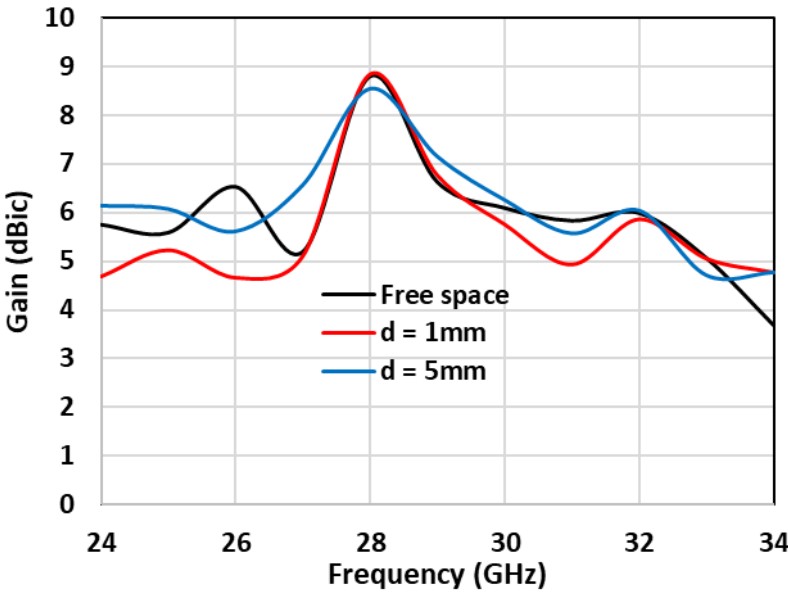

**Figure 5.** Gain of a DRA placed in free space and next to a phantom.

### 2.3. 2 × 2 DRA-Array Design

The presented single-DRA element was utilized to design a compact broadside 4-element array that was positioned on a square ground plane with a size of 36.5 mm, on which the cross-slots were etched, as demonstrated in Figure 6a. A square-array geometry was chosen, as illustrated in Figure 6b, since it offers a more compact structure. The feed network is presented in Figure 6c, where it can be observed that a four-way microstrip power-divider line was utilized for the array-feed network [28]. The dimensions were chosen to support the broadside-array-design criteria, including uniform power division and phase distribution. In addition, two microstrip line widths were chosen, $w_{t1}$ = 0.5 mm and $w_{t2}$ = 0.25 mm, for characteristic impedances of 50 $\Omega$ and 100 $\Omega$, respectively. Furthermore, the lengths of the microstrip line sections were optimized as $l_{t1}$ = 18.13 mm, $l_{t2}$ = 2.5 mm, $l_{t3}$ = 5 mm and $l_{t4}$ = 5.3 mm in order to achieve the widest impedance bandwidth. It should be noted that a Rogers substrate, RO4350B, was employed in the simulations based on commercial availability.

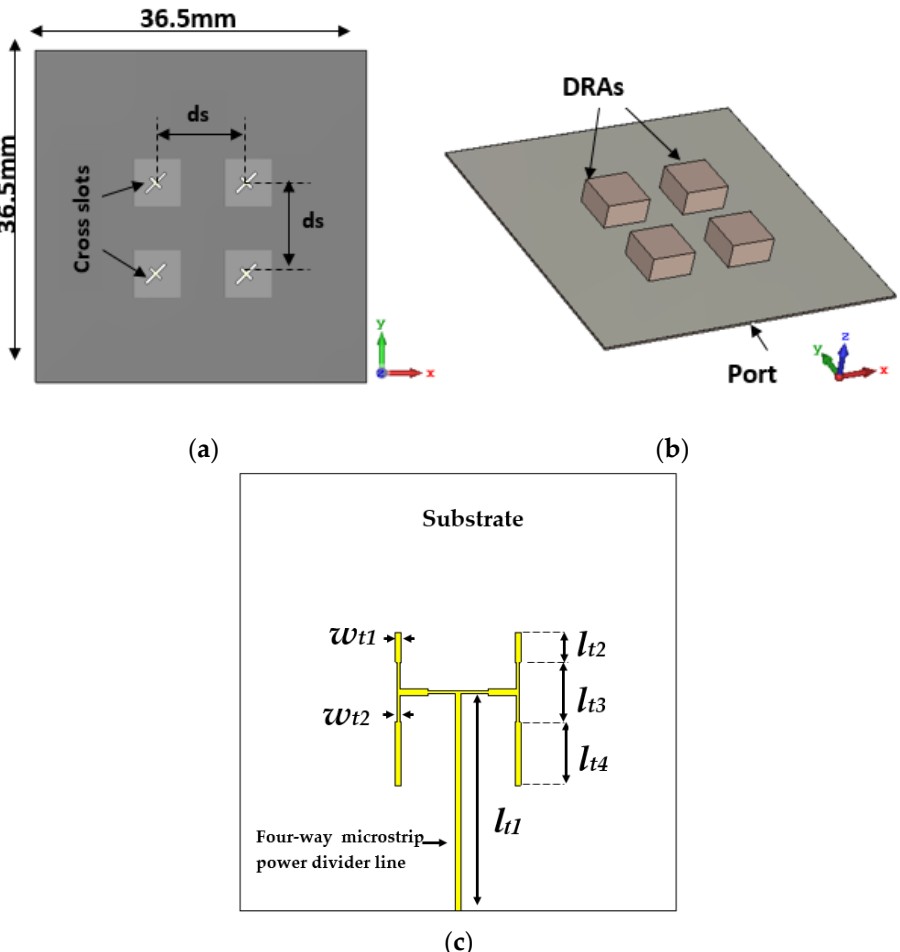

**Figure 6.** Configuration of the proposed DRA-array antenna (**a**) top view, (**b**) three-dimensional view and (**c**) feed network.

The impact of the separation distance, $d_s$, between the centers of the adjacent DRA elements was investigated, as demonstrated in Figure 7. For this investigation, $d_s$ was varied from 6 to 10 mm, which corresponds to a range of $0.558\lambda_o$ to $0.93\lambda_o$, where $\lambda_o$ is the free-space wavelength at 28 GHz. As demonstrated in Figure 7, the widest impedance bandwidth was achieved when the antenna elements were separated by 10 mm, which can be understood as a result of the reduced mutual coupling at this distance. It is noteworthy that the widest recorded impedance bandwidth for the array was 28%, which was close to that of the single DRA element. Moreover, Figures 8 and 9 present the axial ratio and gain as functions of $d_s$, where the widest AR bandwidth of 13% and highest gain of ~14 dBic were achieved, again, when $d_s$ = 10 mm due to the reduced mutual coupling.

### 2.4. Array Next to a Human Body

To learn more about the effects of the human body on the array performance, the same procedure used for a single DRA element was followed, i.e., the DRA array was placed at distances of 1 cm and 5 cm away from the three-layer human-body model, as illustrated in Figure 10.

Figure 11 presents the reflection coefficient when the array is placed in free space and at distances of 1 mm and 5 mm from the phantom. These results confirm that the attained impedance bandwidth is approximately 28% at the two considered distances, which is comparable to that of a DRA array operating in free space. Owing to the presence of a ground plane between the antenna and the simulated tissue layers, the effect of the human tissue on the impedance bandwidth was minimized.

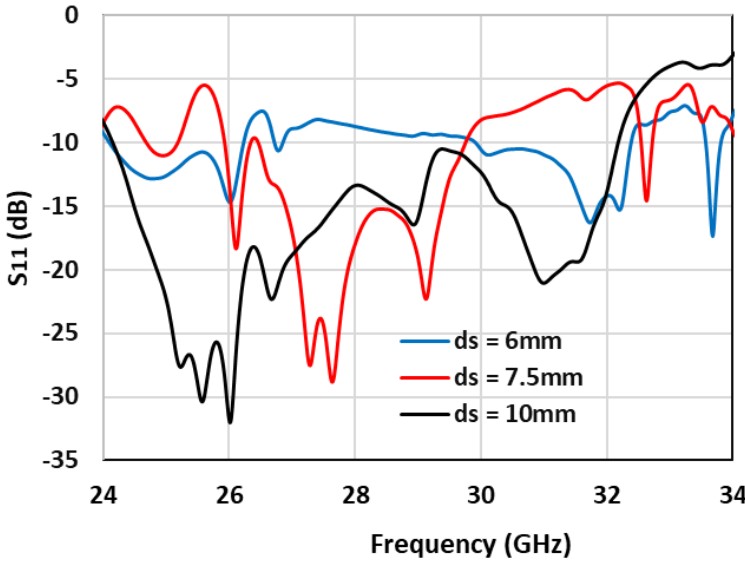

**Figure 7.** $|S_{11}|$ of the DRA arrays for various separation distances ($d_s$).

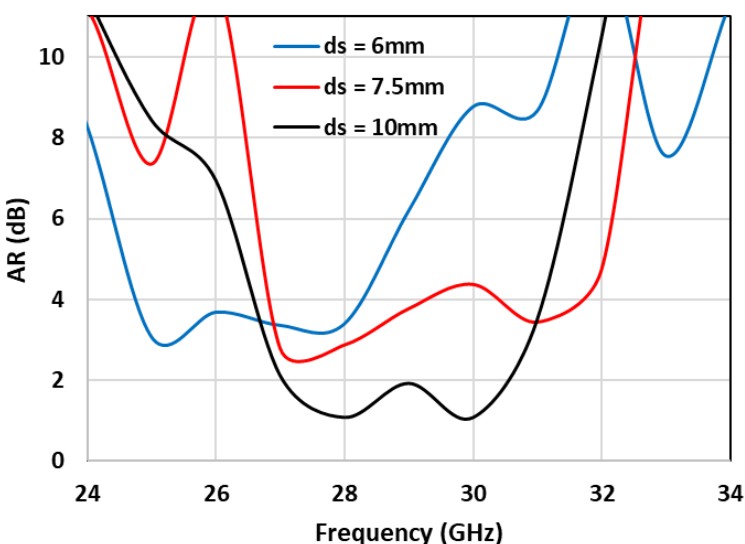

**Figure 8.** Axial ratio of the DRA array using various separation distances ($d_s$) between the DRAs.

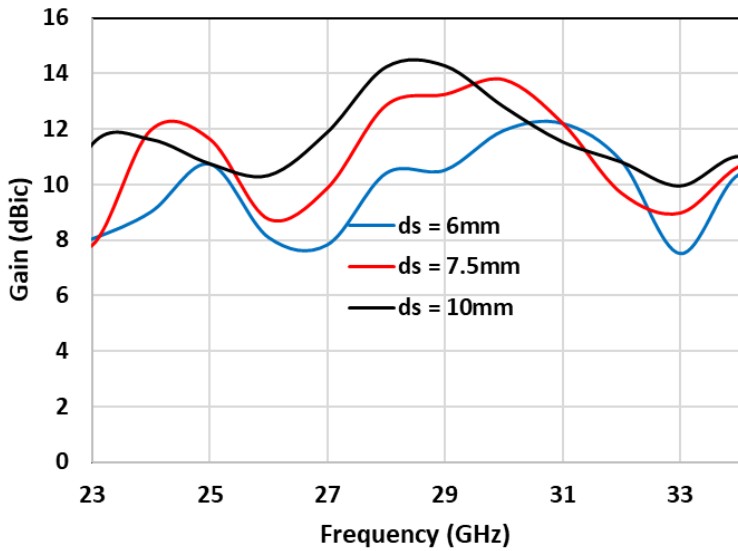

**Figure 9.** Gain of the DRA array using various separation distances ($d_s$) between the DRAs.

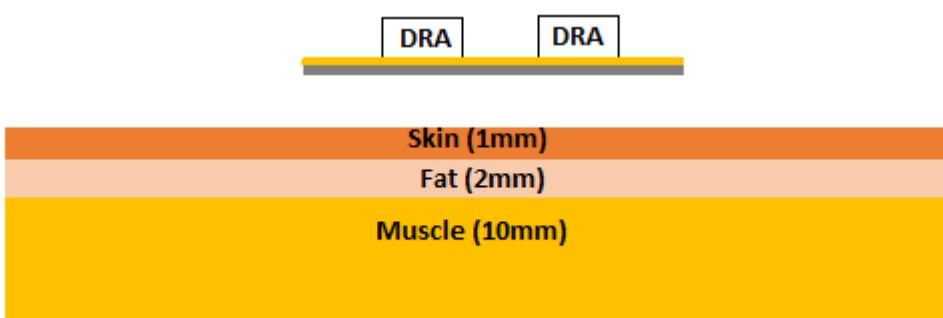

**Figure 10.** Cross-section view of a DRA array next to the body phantom.

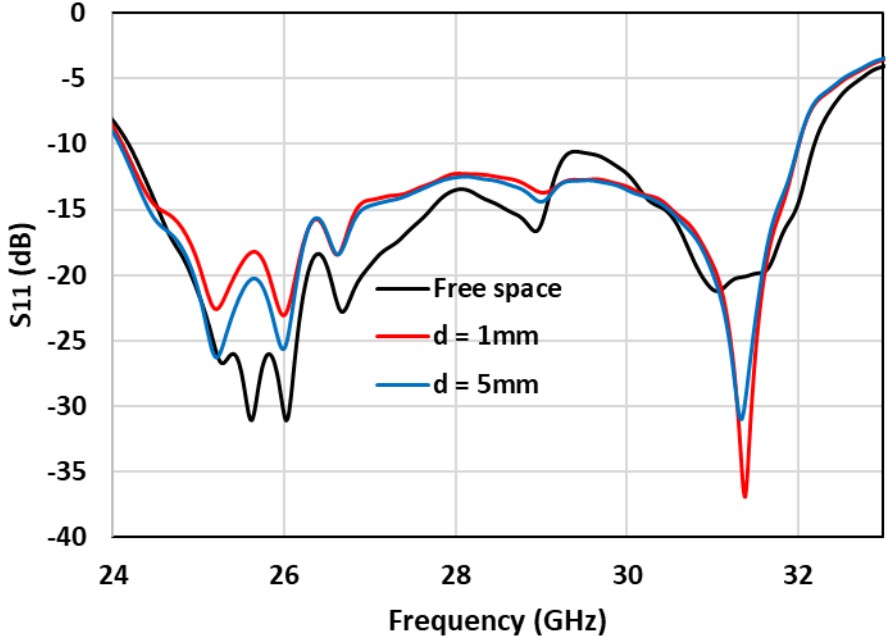

**Figure 11.** Reflection coefficient of a DRA array in free space and placed next to a phantom.

Similarly, the far-field characteristics are not affected by the presence of the human-tissue phantom layers, as demonstrated in Figures 12 and 13, where it is evident that the achieved axial ratio bandwidth of 13% and gain of 14 dBic when the array is located at free space, are comparable to those when the array operates close to the human-tissue model. As mentioned above, the chosen DRA element supports the following TE resonance modes across the considered frequency range of 24–31 GHz: $TE_{111}$, $TE_{113}$ and $TE_{133}$. The excitation of the higher-order modes provided a maximum gain of ~14 dBic at 28 GHz, which is considerably higher than those reported in the literature for four-element DRA arrays. This was to be expected, since the gain of the designed single dielectric-resonator antenna is ~9 dBic, compared to a typical gain of ~6 dBi for a DRA that operates in a lower-order mode. In addition, all the excited modes support broadside radiation, which provided the required stability of the radiation patterns over the considered frequency range. Furthermore, merging the bandwidths of the excited adjacent resonance modes provided considerably wider impedance and CP bandwidths. In addition, the surface-current distribution at the resonant frequency of 28 GHz is depicted in Figure 14, where a clockwise rotation of the current vectors can be observed, which produces the left-hand circularly polarized radiation. It should be noted that the same current distribution was observed for the single circularly polarized DRA, which demonstrates the unnoticeable impact of the mutual coupling between the array elements on the current distribution.

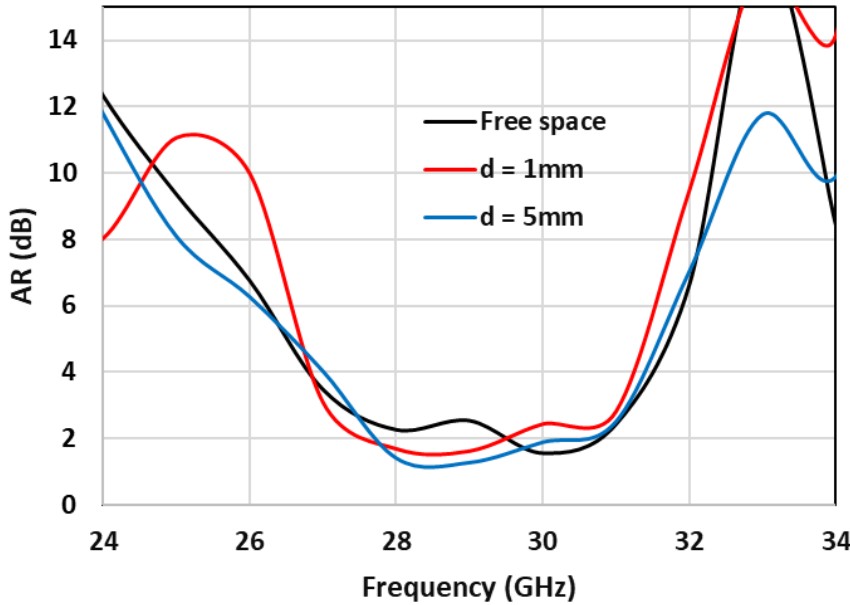

**Figure 12.** Axial ratio of a DRA array placed in free space and next to a body phantom.

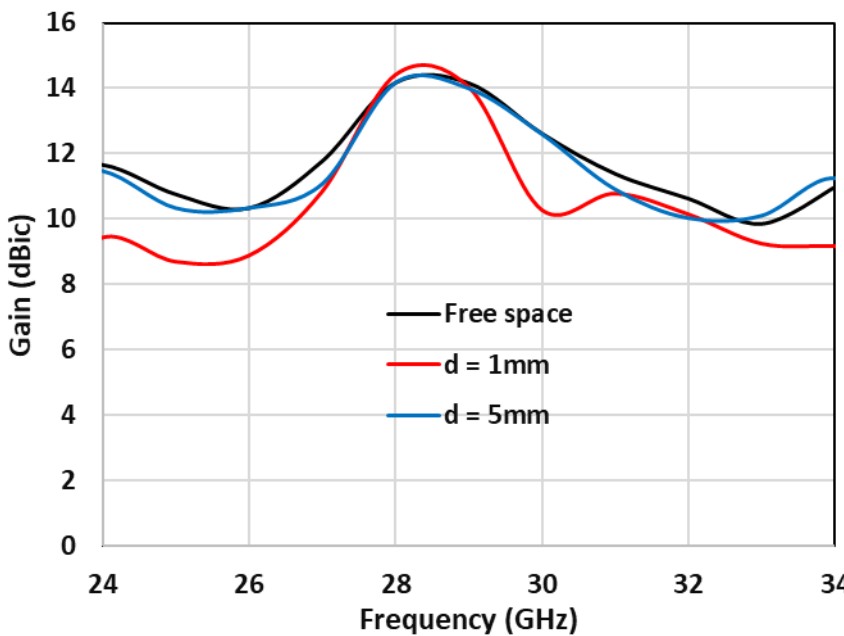

**Figure 13.** Gain of a DRA array operating in free space and next to a body phantom.

*2.5. Specific Absorption Rate (SAR) of the DRA Array*

The SAR represents the rate at which radio-frequency energy is absorbed by human-body tissue [26], which determines the absorbed energy per mass unit according to the following equation [29].

$$\text{SAR} = \sigma \, \frac{\text{E}_i{}^2}{\rho} \ \ \text{W/kg}$$

where $\text{E}_i$ is the strength of the incident electric field, $\rho$ is the tissue's mass density in kg/m$^3$ and $\sigma$ is the tissue's conductivity in S/m. The specific absorption rate needs to be evaluated to ensure that it is lower than the recommended safety-threshold limit according to the Federal Communications Commission (FCC) and International Commission on Nonionizing Radiation Protection (ICNIRP) standards. The respective SAR thresholds are 1.6 W/kg for any 1 g of tissue and 2 W/kg for any 10 g of tissue, as defined by the FCC and ICNIRP

standards [28]. However, it is important to note that these SAR guidelines are defined for frequencies of up to 10 GHz and 6 GHz for the FCC and ICNIRP, respectively, and have not yet been released for higher frequencies. Since near-field exposure to mmWaves has only became a concern in recent years, these guidelines do not offer any dosimetric information or suggestions that could be applied to near-field exposure to mmWaves [26,29,30]. However, it is still important to calculate the SAR, since the antenna operates next to the human body. According to [29], at 28 GHz, the used space between the antenna and the human body can be maintained at 5 mm with the recommended input powers of 15 dBm, 18 dBm, or 20 dBm. Hence, the proposed array was simulated next to equivalent tissues of the skin, fat and muscle, with the characteristics listed in Table 3.

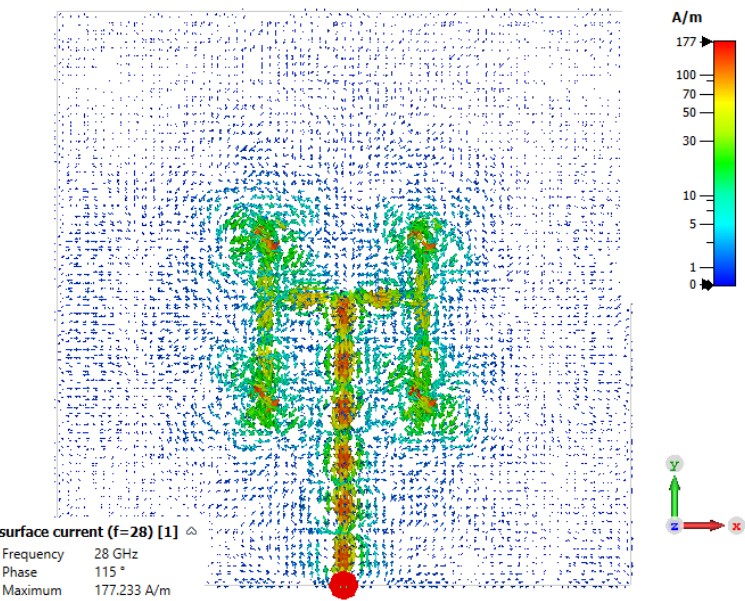

**Figure 14.** Surface-current distribution on the DRA array's ground plane at 28 GHz.

Figures 15 and 16 illustrate the SAR when the proposed DRA array is placed next to the equivalent three-layer tissue of the human body, where it can be noted that the SAR is well below the required safety threshold for different input-power levels when 1-gram and 10-gram tissues are considered. Table 4 summarizes the maximum achieved SAR levels to confirm that the recommended safety levels were maintained according to the FCC and ICNIRP standards.

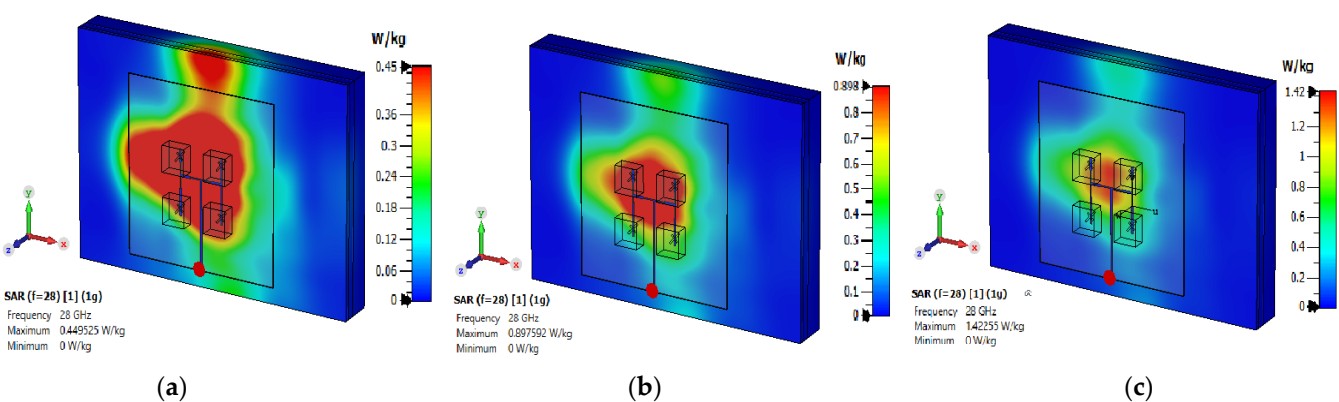

**Figure 15.** The SAR of the DRA array for a 1-gram tissue and input power of (**a**) 15 dBm, (**b**) 18 dBm, (**c**) 20 dBm.

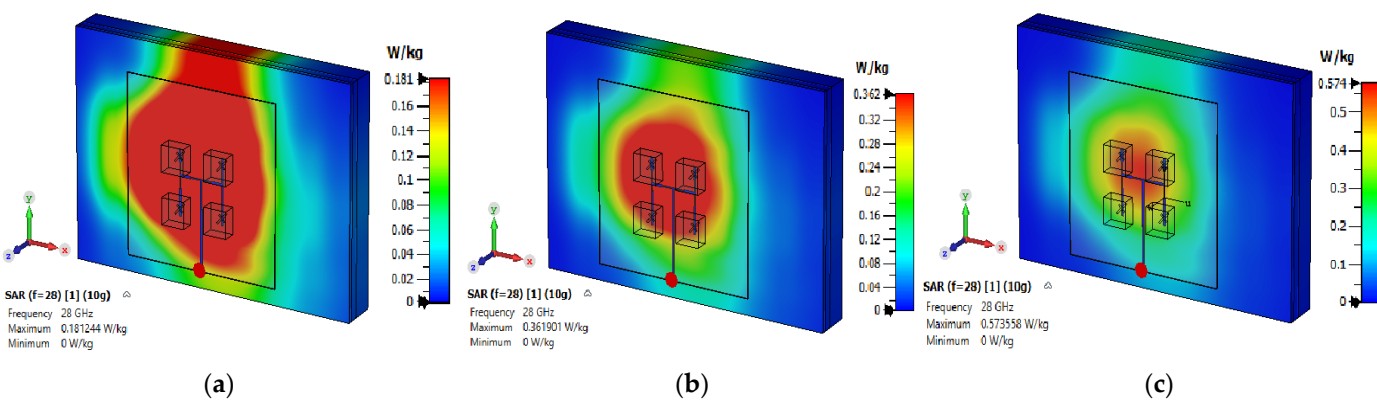

**Figure 16.** The SAR of the DRA array for a 10-gram tissue and input power of (**a**) 15 dBm, (**b**) 18 dBm, (**c**) 20 dBm.

**Table 4.** The average SAR for 1-gram and 10-gram tissues using input power of 15 dBm, 18 dBm and 20 dBm.

| Standard | Input Power (dBm) | SAR(W/kg) | Distance (mm) |
|----------|-------------------|-----------|---------------|
| FCC/ANSI | 15 | 0.4495 | 5 |
| ICNIRP | 15 | 0.18123 | 5 |
| FCC/ANSI | 18 | 0.8975 | 5 |
| ICNIRP | 18 | 0.3619 | 5 |
| FCC/ANSI | 20 | 1.4225 | 5 |
| ICNIRP | 20 | 0.5735 | 5 |

## 3. Measurements

At mmWave frequencies, the precise alignment and assembly of the DRA represent significant challenges that were addressed by outlining the DRA positions on the ground plane. The fabricated feed network is presented in Figure 17a,b, along with the outlined DRA positions. Once the DRA positions were outlined, the antennas were bound to the ground plane using an extremely thin double-sided adhesive copper tape with a thickness of 0.08 mm. The assembled DRA-array prototype is presented in Figure 17c; it was measured without any alignment or bonding concerns.

As with the previous investigation, the human body was observed to have a negligible impact on the performance; therefore, all the measurements were conducted for the DRA array in free space. The NSI-MI Technologies system was utilized for the far-field measurements, and the N5245B vector network analyzer (VNA) was used to quantify the return losses [31]. The measurement system is presented in Figure 17d. Figure 18 demonstrates the simulations and measurements for the array's reflection coefficient, where it can be noted that the measured −10 dB bandwidth was 8.16 GHz. In addition, a good agreement was achieved between the measured and simulated impedance bandwidths of 29% and 28.4%, respectively. However, there is a slight difference in the curves, which may have been the result of marginal fabrication and experimental errors.

The axial ratio and gain of the DRA array are presented in Figures 19 and 20, with close agreement between the measurements and simulations. The measured gain, however, was marginally reduced to 13.7 dBic compared to the maximum simulated gain of 13.9 dBic at 28 GHz. This may be attributed to experimental errors, as well as any slight variations in the dielectric constant of the utilized materials. On the other hand, the measured AR bandwidth was ~13%, which is similar to the simulated counterpart. It should be noted that the achieved AR bandwidth was at least 160% wider than those reported in the literature for DRA arrays operating in the same frequency range [5,20].

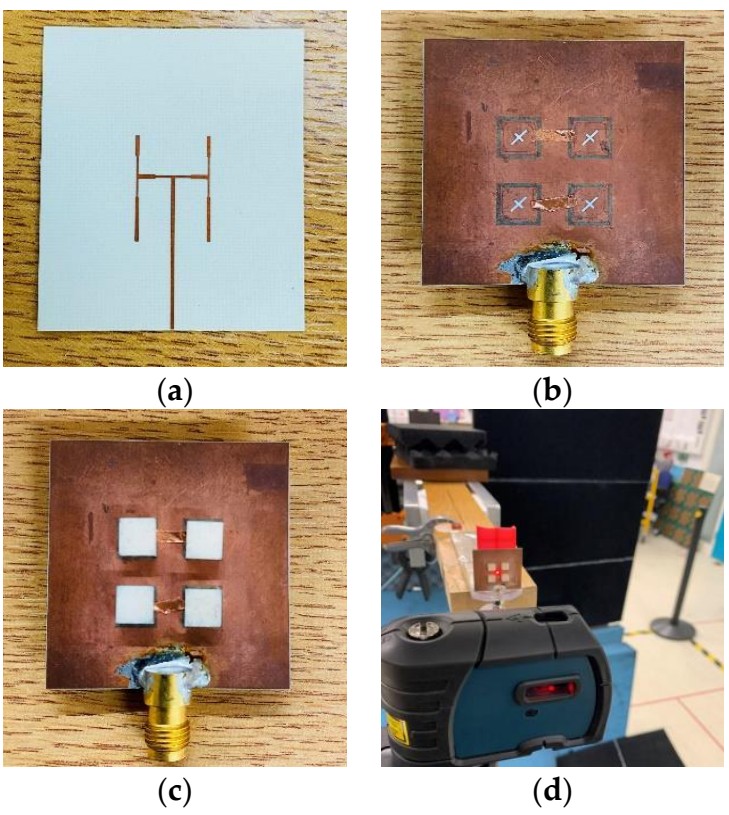

**Figure 17.** The array prototype: (**a**) back view of the feed network, (**b**) top view of the feed network, (**c**) assembled DRA array, (**d**) measurements system.

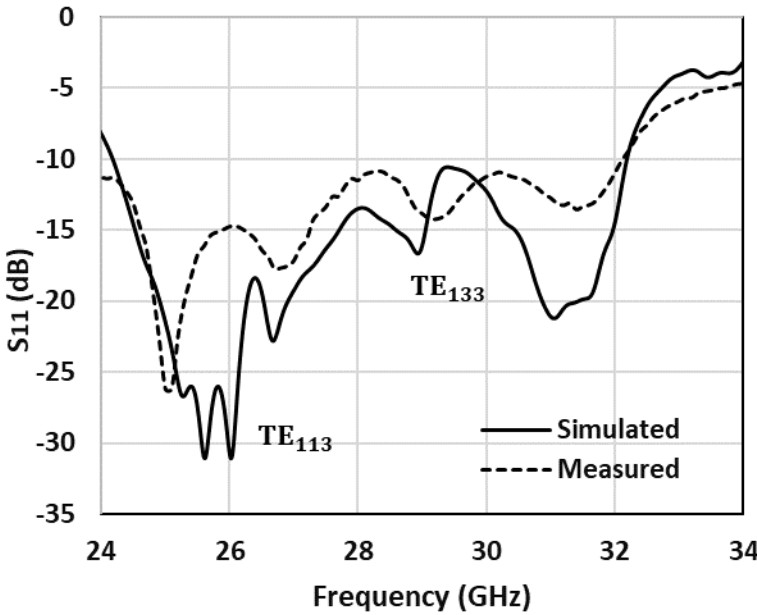

**Figure 18.** The simulated and measured |$S_{11}$| of the DRA array.

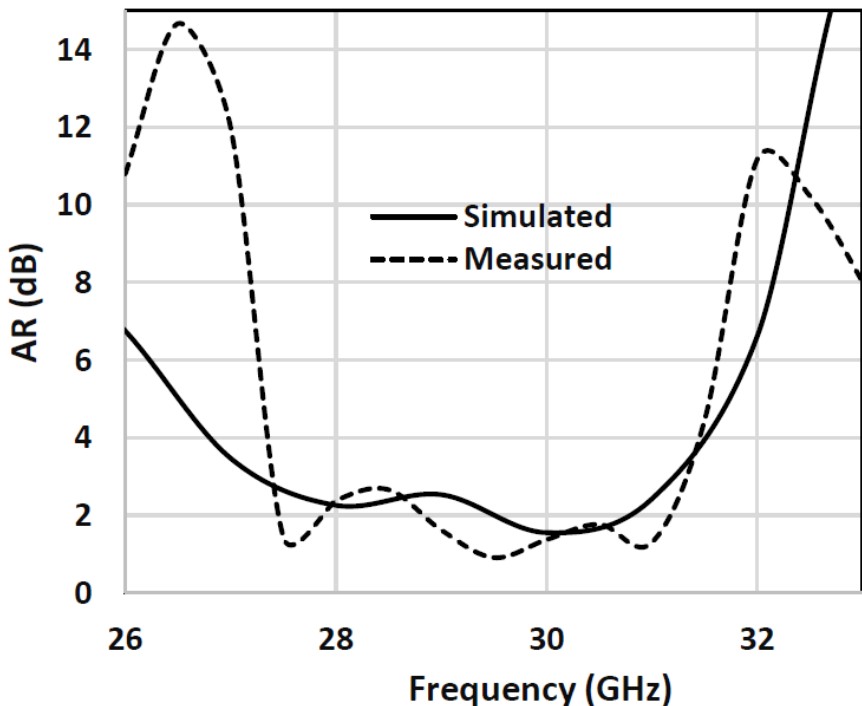

**Figure 19.** The simulated and measured axial ratio of the DRA array.

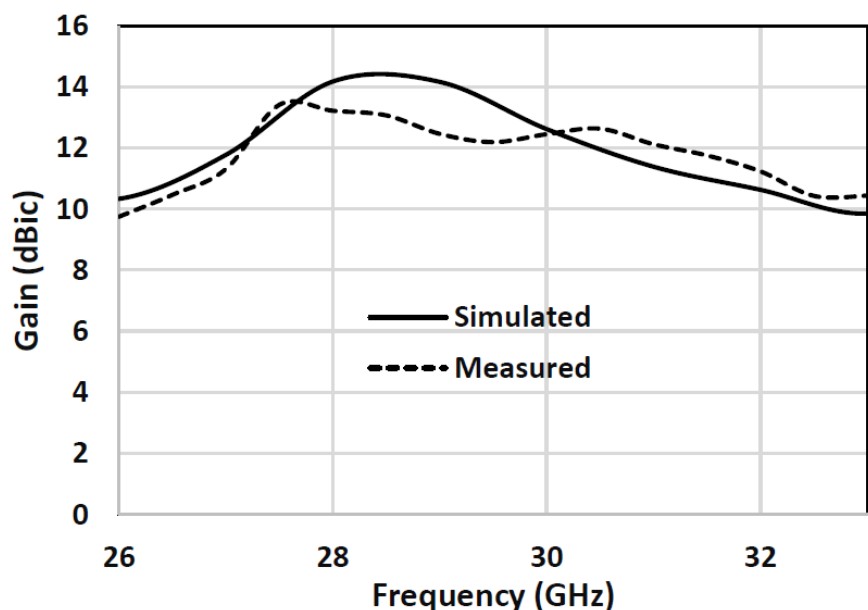

**Figure 20.** The simulated and measured gain of the DRA array.

Figure 21 presents the simulated and measured E- and H-plane radiation patterns at 28 GHz, 29 GHz and 30 GHz for the proposed antenna array in free space. Overall, a good level of agreement can be observed between the measurements and simulations at these frequencies. For example, the respective measured and simulated gains are 13.7 dBic, 13.9 dBic at 28 GHz, 13.68 dBic, 14 dBic at 29 GHz and 12.45 dBic, 12.6 dBic at 30 GHz.

The presented 2 × 2 DRA array's performance parameters, gain, AR, size and bandwidth, are listed in Table 5 and compared to those of various mmWave DRA-array configurations that have been reported in the literature. It can be observed that the proposed array, with only four DRA elements, provides a higher gain than that reported the literature; it even outperforms that achieved using eight elements [19] and offers only 2 dB less than that achieved using sixteen elements [21], which illustrates its higher performance with reduced

complexity. Compared to the circularly polarized designs in [5,17,20], this work provides a wider AR bandwidth combined with higher gain performance. Other designs [18,19,21,22] proposed LP-antenna arrays. Therefore, this work offers improved performance, as shown by the measured gain of 13.7 dBic, in combination with impedance and AR bandwidths of 28% and 13%, respectively, all overserved for a four-element DRA array only.

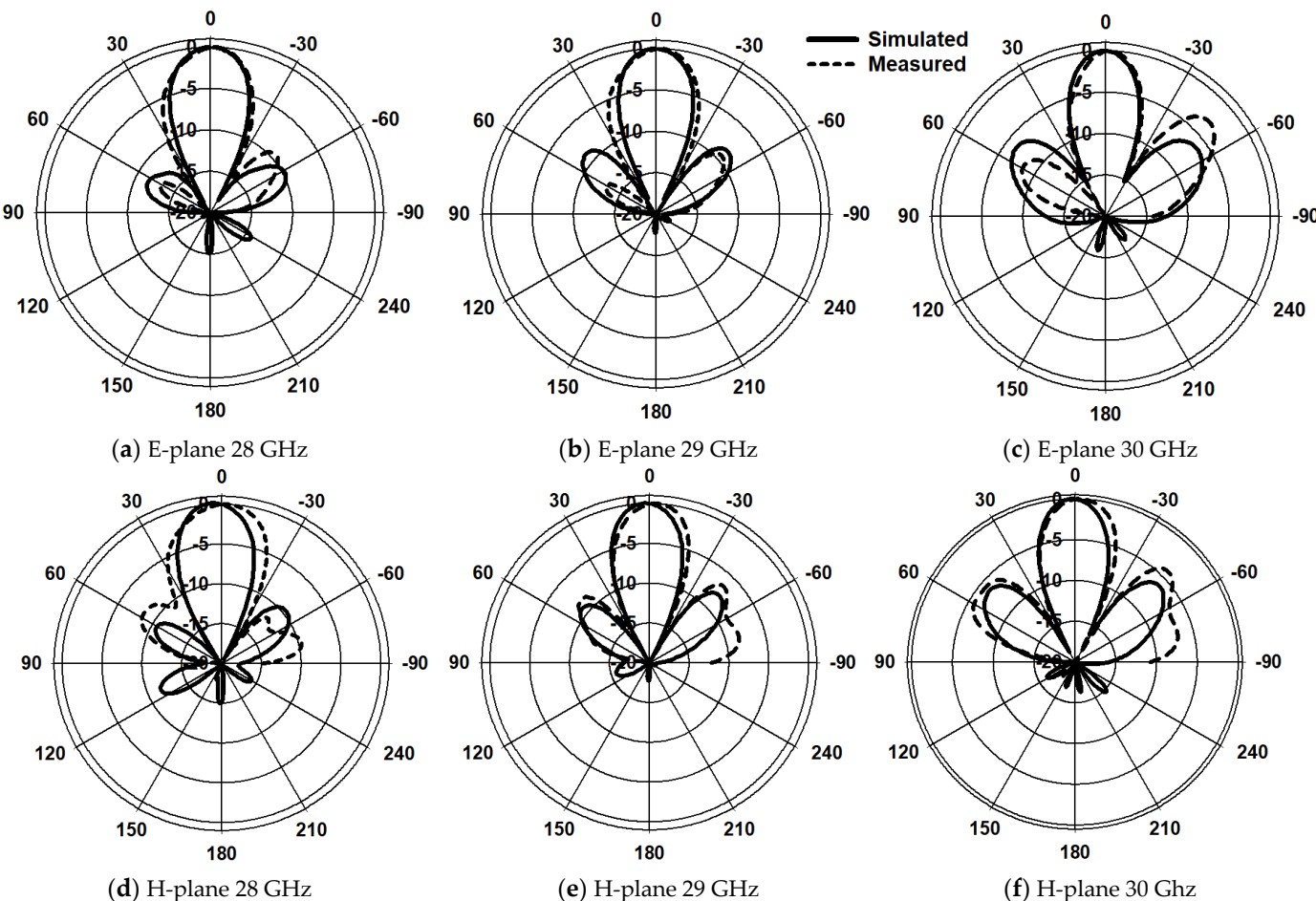

(**a**) E-plane 28 GHz     (**b**) E-plane 29 GHz     (**c**) E-plane 30 GHz

(**d**) H-plane 28 GHz     (**e**) H-plane 29 GHz     (**f**) H-plane 30 Ghz

**Figure 21.** Simulated and measured radiation patterns across the CP bandwidth of the DRA array.

**Table 5.** Comparison of the proposed antenna performance with those of previously published designs.

| Ref. | Elements Number | $f_c$ (GHz) | Gain | $S_{11}$ BW (%) | AR BW (%) | Size (mm³) |
|---|---|---|---|---|---|---|
| [5] | 4 | 30 | 12.7 dBic | 16.4 | 1.1 | - |
| [17] | 4 | 60 | 11.4 dBic | 26 | 15.9 | - |
| [18] | 4 | 60 | 10.5 dBi | 12 | LP | - |
| [19] | 8 | 32 | 12 dBi | 18.29 | LP | - |
| [20] | 4 | 30 | 9.5 dBic | 33.8 | 5 | 20 × 20 × 1.52 |
| [21] | 16 | 28.7 | 15.68 dBi | 9.81 | LP | 46 × 46 × 1.5 |
| [22] | 4 | 27.5 | 9.8 dBi | 31.6 | LP | 47 × 8 × 1.084 |
| This work | 4 | 28 | 13.7 dBic | 29 | 13 | 36.5 × 36.5 × 2.9 |

## 4. Conclusions

This study presented a four-element circularly polarized rectangular DRA array for mmWave off-body applications. For the suggested antenna, cross-slot coupling, together

with a sequential parallel feeding network, was utilized to achieve a wide impedance bandwidth of 29% while maintaining a high radiation efficiency, of 90%. The DRA offers a CP-bandwidth improvement of more than 160% compared to reported counterparts that operate at the same frequency range. In addition, exciting a higher-order resonance mode provided a notably higher gain over a frequency range of 26–31 GHz. The SAR was investigated and it was demonstrated that the proposed array meets the required safety limits when placed at a distance of 5 mm from the human-body tissue. The proposed antenna array was fabricated and measured with a gain of 13.7 dBic, which represents an enhancement of 5 dBi above that of a single DRA. The impact of human-body proximity was investigated and found to be negligible due to the presence of the ground plane. The results of simulations and measurements are in close agreement. The proposed antenna outperforms the mmWave DRA arrays reported previously in the literature in terms of gain and CP bandwidth and offers promising potential for mmWave off-body communications.

**Author Contributions:** T.S.A.: simulation, manufacturing and measurements, writing; R.S.: writing, S.K.K.: supervision and writing. All authors have read and agreed to the published version of the manuscript.

**Funding:** This research received no external funding.

**Institutional Review Board Statement:** Not applicable.

**Informed Consent Statement:** Not applicable.

**Data Availability Statement:** Not applicable.

**Acknowledgments:** The authors would like to acknowledge the use of the National mmWave Measurement Facility and thank Steve Marsden for his support with the measurements. Furthermore, the authors would like to thank Matthew Davies for outlining the DRA positions on the feed network.

**Conflicts of Interest:** The authors declare no conflict of interest.

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
