# Peer review of "A Circularly Polarized mmWave Dielectric-Resonator-Antenna Array for Off-Body Communications"

_applsci, doi:10.3390/app13032002_

Round 1
Reviewer 1 Report
The authors presented a Circularly Polarized mmWave Dielectric Resonator Antenna Array for Off-Body Communications. The concept is exciting, and the simulation results are reasonably good, showing potentially strong reconfigurability. I have the following suggestions before accepting it for publication:
- In the introduction part, the author must discuss the techniques for circular polarised printed antenna systems [1,2].
[1] Broadband Circular Polarised Printed Antennas for Indoor Wireless Communication Systems: A Comprehensive Review. Micromachines 2022, 13, 1048. https://doi.org/10.3390/mi13071048.
[2] A Compact Sequentially Rotated Circularly Polarized Dielectric Resonator Antenna Array. Appl. Sci. 2021, 11, 8779. https://doi.org/10.3390/app11188779.
- Since the antenna is applied for wearable applications, I want to see the Specific Absorption Rate (SAR) calculation weights in terms of 1 gram up to 10 grams of tissue?
- It needs to be evident how the proposed design can be used to address the research gaps of the current study. Significant improvements need to be made by the authors to emphasize the significance of this paper. For example, how many percentages of Axial Ratio improvements are obtained by the proposed design?
- What impact does the dielectric resonator antenna (DRA) in the proposed wearable antenna have on the circularly polarized technique at mmWave frequencies?
- The quality of the figures presented in this paper is relatively poor, and the fonts are too small to read. Please improve the quality of the figures.
- The differences between the simulated results and the measured results in Figure 12 should be discussed more.
- The Conclusions should be rewritten with the updated results above.
The authors are required to revise the comments mentioned above carefully.
Author Response
Dear Sir/Madam
Many thanks for the constructive comments, please find our responses in the attached file.
Regards
Mr Tarek Abdou

Reviewer 2 Report
The authors have presented the manuscript titled “A Circularly Polarized mmWave Dielectric Resonator Antenna 2 Array for Off-Body Communications”. My comments are as follows:
1. Mention the dimension of the antenna in terms of electrical length in the abstract.
2. Quality of all the figures should be improved
3. In Fig. 2, the height of skin, muscle and fat should be given individually and not combined as 13mm.
4. Also mention the properties of all the tissues at the resonating frequency used to design the body phantom in a separate table
5. In Figure 4, the scale of Axial ratio is missing. Also, it is difficult to analyze the curves of gain and Axial ratio, hence authors should use different colors curve for axial ratio and different for gain or can plot a separate graph. Same with Figures 7, 10 and 13.
6. The surface current distribution should be added.
7. The authors should include vector current distribution to prove circular polarization of the antenna, the authors can refer the below manuscripts and cite them:
a. Dual Circularly Polarized Planar Four-Port MIMO Antenna with Wide Axial-Ratio Bandwidth
b. Broadband and Compact Circularly Polarized MIMO Antenna With Concentric Rings and Oval Slots for 5G Application
c. A novel millimeter-wave dual-band circularly polarized dielectric resonator antenna
d. Dual Polarized, Multiband Four-Port Decagon Shaped Flexible MIMO Antenna for Next Generation Wireless Applications
Author Response

(The authors gave the same response as above.)

Round 2
Reviewer 1 Report
The authors have revised the given comments successfully, and I believe the article is ready now to be published in a reputational journal like Applied Sciences. However, there are still typos and spacing errors that need to be carefully checked.
Author Response
Dear Sir/Madam
Many thanks for your constructive comments.
Kind regards
Mr Tarek Abdou

Reviewer 2 Report
All the comments are addressed carefully. Kindly cite the remaining two references suggested by a reviewer as follows:
A novel millimeter-wave dual-band circularly polarized dielectric resonator antenna
Dual Polarized, Multiband Four-Port Decagon Shaped Flexible MIMO Antenna for Next Generation Wireless Applications
Author Response

(The authors gave the same response as above.)
